# High Rate of Discontinuation during Long-Acting Injectable Antipsychotic Treatment in Patients with Psychotic Disorders

**DOI:** 10.3390/biomedicines11020314

**Published:** 2023-01-22

**Authors:** Anna Maria Auxilia, Massimiliano Buoli, Alice Caldiroli, Greta Silvia Carnevali, Agnese Tringali, Roberto Nava, Massimo Clerici, Enrico Capuzzi

**Affiliations:** 1School of Medicine and Surgery, University of Milano Bicocca, via Cadore 38, 20900 Monza, Italy; 2Department of Neurosciences and Mental Health, Fondazione IRCCS Ca’ Granda Ospedale Maggiore Policlinico, 20122 Milan, Italy; 3Department of Pathophysiology and Transplantation, University of Milan, 20122 Milan, Italy; 4Psychiatry, Fondazione IRCCS San Gerardo dei Tintori, 20900 Monza, Italy

**Keywords:** long acting injectable, discontinuation, psychotic disorders, long-term treatment, clinical practice, schizophrenia, bipolar disorder, first-generation antipsychotics, second-generation-antipsychotics, retrospective study

## Abstract

Treatment discontinuation is a major challenge in routine clinical settings. Despite poor adherence to antipsychotic medication, long acting injectable (LAI) formulations are an underutilized option in psychotic disorders. Recently, an earlier and broader use of LAIs has been emphasized. However, few studies have evaluated the factors associated with LAI antipsychotic discontinuation in ordinary clinical practice. The main purpose of the present study was, therefore, to identify the factors associated with LAI discontinuation in a real-world setting. Patients in treatment with LAI antipsychotics were recruited. A Cox regression analysis was applied considering a 12-month follow-up period. Moreover, a Kaplan-Meier survival analysis was applied to compare the single treatment LAI antipsychotic groups in terms of time to discontinuation. Our analysis showed an LAI discontinuation rate at 12 months, corresponding to 28.8%, with olanzapine and aripiprazole having a longer time to discontinuation compared to zuclopenthixol. The results of the present study can help clinicians with their choice of LAI antipsychotic according to patients’ characteristics and in a context of precision medicine. Increasing knowledge about factors affecting discontinuation of LAI antipsychotics can improve the prescribing practices of these compounds. Individualized approaches may ameliorate long-term patients’ treatment adherence, thus preventing the long-term disability caused by psychotic disorders.

## 1. Introduction

The progressive aging of the world population, particularly in Western countries, is associated with an increase of the prevalence of different chronic conditions. Chronic diseases are generally defined as physical or mental disorders that last more than one year, requiring ongoing care [1]. Chronic disease can compromise an individuals’ physical and social function, their quality of life as well as the economic sustainability of healthcare systems. Therefore, the management of these conditions is fundamental in order to minimize their impact on patients’ wellbeing and optimize healthcare costs [2]. Patients’ treatment adherence represents an important factor in the successful management of chronic illnesses. Of note, only half of patients with chronic diseases take their medications as prescribed, making medication compliance a priority of the public health agenda [3]. Schizophrenia-spectrum disorders and other psychotic disorders are chronic conditions, representing leading causes of disability worldwide. Indeed, more than 50% of individuals with psychotic disorders has re-exacerbation of the illness and up to 20% have chronic symptoms and disability [4]. The problem of non-adherence to medications is a major issue in people with psychotic disorders. Although, generally, these conditions require a continuative pharmacological treatment in order to avoid relapses, it has been estimated that approximately half of patients have already interrupted their maintenance therapy six months after the first administration [5]. Indeed, there is a number of factors that hamper treatment compliance, including lack of insight or illness-related cognitive impairment, especially for subjects affected by schizophrenia [6]. However, it may be difficult to identify at-risk subjects or those who discontinued treatment in clinical practice, since both patients and clinicians tend to overestimate compliance [7]. Poor compliance to pharmacotherapy worsens the course of psychiatric disorders and accounts for social and economic burden [8]: of note, patients with scarce adherence to medical prescriptions exhibit an increased risk of relapses and rehospitalizations, associated with work loss and social impairment [9].

The psychopharmacological treatment of schizophrenia and other psychotic disorders includes the use of antipsychotics, which are traditionally divided into two classes: first generation antipsychotics (FGA) and second-generation antipsychotics (SGA). Both classes of drugs are effective in treating positive symptoms (delusions and hallucinations), whilst data about the efficacy on negative, cognitive and affective dimensions are contrasting and FGAs may even worsen these symptoms. Nevertheless, antipsychotics may also induce different types of side-effects [5]. Current guidelines recommend maintenance treatment with antipsychotic drugs after the first psychotic episode, while intermittent treatment is discouraged [10]. Nevertheless, long-acting formulations of antipsychotics are commonly prescribed in chronic patients with poor adherence to oral treatment, favoring compliance [11]. On the other hand, several studies demonstrated the efficacy of LAI compounds since the first episode of illness, because, similarly to clozapine, these formulations can ensure a prolonged remission, functional recovery and reduced risk of disorder progression [12]. LAI antipsychotics proved an 85% reduction of relapse risk in patients with a first psychotic episode, as well as a 20–30% decrease of rehospitalization risk compared to an oral therapy with the same compound [11]. Furthermore, LAI treatment avoids first-pass metabolism in the liver with less fluctuations of medication plasma levels, which is, in turn, associated with more clinical stabilization [4]. Another beneficial impact of LAIs is the improvement of family relationships, as the caregivers are relieved of the need to check on patients’ medication compliance [13]. As mentioned above, the literature seems to agree that an adequate adherence to treatment leads to a reduction in direct and indirect costs, especially for patients affected by schizophrenia, but also as concerns those with bipolar disorder, although the use of LAI antipsychotics has not yet been approved by EMA for bipolar disorder [12].

Drop-out rates from LAI antipsychotics are observed in the real world. However, current data show that discontinuation rates of second generation antipsychotic LAIs (SGA-LAIs) seem similar to those of first-generation compounds (FGA-LAIs) [14], although a study found that haloperidol decanoate had the lowest discontinuation rate [15]. With regard to aripiprazole and olanzapine, the dropout rate might go up to 50%, while for risperidone, it was lower when compared to both oral treatment and FGA-LAIs [16]. However, according to another research, only 20% of patients taking risperidone-LAI continued the therapy for more than 90 days [17]. Some authors suggested that adverse events (particularly sexual, metabolic and extrapyramidal effects), inadequate efficacy of medications, severity of symptoms, in particular related to the domains of suspiciousness/hostility, attitudes toward medication, educational level, cognitive impairment, a concomitant substance use disorder, poor insight contribute to poor adherence to LAI antipsychotics [16,18,19]. Nevertheless, another important factor might be represented by the phase of the disorder, since first episodes are generally associated with a good response to treatment: an asymptomatic individual who achieved stable remission might not understand the necessity to maintain the therapy [20]. In light of these considerations, integrated and personalized interventions, including behavioral and psychoeducational approaches, are useful to reduce the chance of treatment discontinuation [21]. However, few data have been published until now about the factors associated with medication discontinuation in patients treated with LAI antipsychotics, also with regard to possible differences between FGA-LAIs and SGA-LAIs.

In this framework, the objectives of the present study are: (1) to identify clinical and demographic predictors of LAI antipsychotic discontinuation during the first year of treatment in patients suffering from psychotic disorders; (2) to investigate potential differences in the time of discontinuation between the LAI compounds (haloperidol, zuclopenthixol, risperidone, paliperidone, aripiprazole, olanzapine); (3) to evaluate other clinical factors (including tolerability) possibly associated with the prescription of a specific LAI antipsychotic. Therefore, using real world data, our main research questions were: (1) What is the rate of LAI antipsychotic discontinuation in a sample of every-day patients with psychotic disorders during a 12 month follow-up period? (2) Are there some potential clinical and demographic predictors of LAI discontinuation in patients with psychotic disorders? (3) Are there some differences in terms of discontinuation between different LAI antipsychotic drugs? (4) Are there some clinical factors associated with the prescription of specific LAI antipsychotics? Therefore, factors associated with LAI antipsychotic discontinuation should be identified in order to implement personalized medicine [22].

## 2. Materials and Methods

Patients with a diagnosis of schizophrenia, schizoaffective disorder or bipolar disorder type I (with lifetime psychotic features) according to DSM (Diagnostic and Statistical Manual of Mental Disorders, American Psychiatric Association, 2013) [23] criteria were enrolled among patients followed up at community mental health services and day hospital afferent to Fondazione IRCCS Policlinico (Milan, Italy) (N = 106) and ASST Monza (N = 276) between January 2007 and July 2021. The study had a retrospective design. Clinical data were obtained through a screening of the clinical charts, Lombardy electronic database for patients affected by mental disorders (SIPRL) and interviews with patients and their relatives [24]. The following inclusion criteria were used to define the sample: age ≥ 18, one of the above-mentioned diagnoses, treatment with a LAI antipsychotic for at least two months. Exclusion criteria consisted of: a treatment with a LAI antipsychotic < 2 months, interruption of treatment for medical conditions that are independent from the administered antipsychotic, interruption of treatment for pregnancy, the presence of intellectual disability, a main diagnosis different from schizophrenia, schizoaffective disorder or bipolar disorder (e.g., very severe personality disorders that need of LAI antipsychotic treatment), an age < 18 years. The protocol of this study was approved by the local Ethical Committee.

The data of the following demographic and clinical variables (referred to the visit before starting LAI) were collected: age, gender, marital status, work status, history of criminal acts, age at onset and duration of illness (years), duration of untreated illness (DUI), presence and number of previous suicide attempts, presence and number of previous hospitalizations, type of LAI medication with mean monthly dosage, presence of personality disorders, family history of psychiatric disorders, multiple family history of psychiatric disorders, pre-onset and post-onset psychiatric comorbidity and poly-comorbidity, pre-onset and post-onset substance misuse and poly-misuse, pre-onset and post-onset medical comorbidity and poly-comorbidity, current poly-pharmacotherapy, current or lifetime psychotherapy, reason for discontinuation of LAI antipsychotic, side effects of therapy and presence of multiple side effects.

Duration of untreated illness was considered as the time elapsing between the onset of the disorder and the prescription of a proper treatment (antipsychotics for schizophrenia, and atypical antipsychotics or mood stabilizers for bipolar disorder) [25,26].

Descriptive statistics on the total sample were performed. A Cox regression analysis was applied considering a 12-month follow-up period. This model was realized to identify the predictors of LAI antipsychotic discontinuation [27]. The event was, therefore, defined as the discontinuation of LAI antipsychotic for recurrences (including those leading to hospitalizations), side effects or lack of compliance [28]. The recurrence was defined as the presence of an acute exacerbation for schizophrenia/schizoaffective disorder or a full mood episode for bipolar disorder according to DSM criteria (American Psychiatry Association, 2013). A survival analysis (Kaplan–Meier) was applied to compare the single treatment LAI antipsychotic groups (haloperidol decanoate, zuclopenthixol decanoate, paliperidone palmitate, olanzapine pamoate, aripiprazole and risperidone) in terms of time to discontinuation in a follow-up period of 12 months [29]. The same analysis was performed grouping the antipsychotics in first generation antipsychotics (haloperidol and zuclopenthixol) and atypical antipsychotics (risperidone, paliperidone, olanzapine, aripiprazole). The treatment groups (single compounds, and neuroleptics versus atypical antipsychotics) were also compared for quantitative variables by multivariate analyses of variance (with post-hoc analyses where appropriate) and for qualitative variables (including side effects and reason of discontinuation) by χ2 tests.

The level of statistical significance was set at *p* ≤ 0.05. All statistical analyses were performed using SPSS (IBM Corp. Released 2020. IBM SPSS Statistics for Windows, Version 27.0. Armonk, NY, USA).

## 3. Results

The total sample consisted of 382 patients: 221 males (57.9%) and 161 females (42.1%) with a mean age of 45.430 years (±12.795). One hundred and one subjects (26.4%) were affected by psychotic bipolar disorder, 59 by schizoaffective disorder (15.5%) and 222 by schizophrenia (58.1%). The mean dose of LAI antipsychotics was: 96.790 ± 40.434 mg/4 weeks for haloperidol (N = 150), 215.800 ± 100.420 mg/4 weeks for zuclopenthixol (N = 44), 111.679 ± 32.859 mg/4 weeks for paliperidone (N = 77), 340.318 ± 79.042 mg/4 weeks (N = 22) for olanzapine, 371.80 ± 50.253 mg/4 weeks (N = 56) for aripiprazole, 39.46 ± 11.656 mg/2 weeks for risperidone. If these doses are converted in olanzapine equivalents: the 4-week dose of zuclopenthixol resulted to be less than that of haloperidol (*p* < 0.01), paliperidone (*p* < 0.01) and olanzapine (*p* < 0.01).

Descriptive analyses of the total sample are reported in Table 1.

The Cox regression (Table 2) showed that baseline type of prescribed LAI antipsychotic predicted long-term therapy discontinuation (*p* = 0.033). A trend to statistical significance was found also for the presence of pre-onset poly-substance use disorders (*p* = 0.058). The results were practically the same when duration of illness was removed from the model (type of treatment *p* = 0.037; pre-onset poly-substance use disorders *p* = 0.051). Proportionality of hazards was satisfied for age (*p* = 0.070) and duration of illness (*p* = 0.444). Patients with pre-onset poly-substance use disorder had a shorter time of discontinuation (8.760 months ± 3.673) compared to the others (10.300 months ± 3.194). With regard to survival analysis, patients treated with olanzapine (Breslow: χ2 = 4.358, *p* = 0.037) and aripiprazole (Breslow: χ2 = 3.877, *p* = 0.049) had a longer time to discontinuation with respect to those treated with zuclopenthixol (Figure 1). No statistically significant differences in terms of time of discontinuation were found between FGA and SGA-LAIs (Breslow: χ2 = 0.378, *p* = 0.539).

The frequency of side effects (moderate in intensity) was different in the treatment groups (χ2 = 143.260, *p* < 0.001). Sedation (N = 8, 36.4%) and sexual dysfunction (N = 3, 13.6%) were more frequently reported in patients treated with olanzapine than the other treatment groups (*p* < 0.05), while motor symptoms emerged more often in patients treated with haloperidol (N = 60, 40%) than in those in treatment with other compounds (*p* < 0.05). This latter result was also confirmed by the analysis comparing FGA versus SGA-LAIs (χ2 = 32.299, *p* < 0.001). Patients reporting side effects were older (F = 5.458, *p* = 0.020), more frequently women (χ^2^ = 7.353, *p* = 0.007), more often treated with poly-therapy (χ^2^ = 12.131, *p* < 0.001), with less frequent pre-onset medical poly-comorbidity (χ^2^ = 5.626, *p* = 0.018) and with less frequent pre-onset poly-substance use disorders (χ^2^ = 4.262, *p* = 0.039) than the counterpart. Duration of illness was not significantly different in the two groups identified by the presence or absence of treatment side effects when co-variating for the type of compound (F = 2.028, *p* = 0.155). Furthermore, patients with poor compliance resulted to have a shorter duration of untreated illness (F = 3.889, *p* = 0.049), a lower number of lifetime suicide attempts (F = 3.956, *p* = 0.047), reported less multiple side effects (χ^2^ = 7.313, *p* = 0.007) and had a more frequent history of criminal acts (χ^2^ = 3.977, *p* = 0.046) than their counterparts. The frequency of prescription of the compounds was different among diagnoses: patients affected by schizophrenia were more likely to receive haloperidol and less likely to receive aripiprazole compared to subjects suffering from schizoaffective or bipolar disorder type I with psychotic features (χ2 = 59.943, *p* < 0.001). This latter result was also confirmed by the analysis comparing FGA versus SGA-LAIs (χ2 = 33.611, *p* < 0.001). The frequency of prescription of the compounds was different among genders (χ2 = 12.358, *p* = 0.030): zuclopenthixol was preferentially prescribed in males compared to females (*p* < 0.05). In addition, patients in treatment with paliperidone or olanzapine showed less pre-onset medical comorbidity than the other treatment groups (χ2 = 14.591, *p* = 0.012). The distribution of treatments was also significantly different according to the lifetime administration of psychotherapy (χ2 = 12.287, *p* = 0.031): patients who received psychotherapy during the course of life were more likely in treatment with aripiprazole or paliperidone (*p* < 0.05). In addition, patients treated with SGA-LAIs were more likely to have received lifetime psychotherapy than those in treatment with FGA-LAIs (χ2 = 11.201, *p* = 0.001). Specifically supportive psychotherapy had been more frequently administered in patients treated with SGA-LAIs than in those in treatment with FGA-LAIs (χ2 = 11.201, *p* = 0.005). In addition, the patients treated with SGAs received psychotherapy more frequently than the others (χ2 = 5.805, *p* = 0.016), while those in treatment with FGAs more frequently had post-onset psychiatric comorbidity than their counterparts (χ2 = 7.094, *p* = 0.008). On the contrary, pre-onset substance use disorders were more frequent in patients treated with atypical antipsychotics than those in treatment with typical ones (χ2 = 4.710, *p* = 0.030). Finally, patients treated with zuclopenthixol had a significantly longer duration of illness with respect to subjects in treatment with aripiprazole (*p* = 0.029), and generally, patients treated with FGA-LAIs had a longer duration of illness than those to whom SGA-LAIs were prescribed (F = 4.327, *p* = 0.038). Patients in treatment with olanzapine showed a longer treatment duration compared to those in treatment with zuclopenthixol (*p* = 0.019) and risperidone (*p* = 0.012), as shown by univariate analysis weighted for duration of illness.

No other significant differences were found between treatment groups (both considering the single compounds and SGA versus FGA-LAIs) with regard to the other variables (*p* < 0.05).

## 4. Discussion

In this naturalistic retrospective study, we investigated the rate of discontinuation of LAI antipsychotics at 12 months in a sample of 382 individuals with schizophrenia, schizoaffective disorder or bipolar disorder, receiving day-hospital or outpatient care. The present research produced four main results. First, LAI antipsychotics are primarily prescribed to people affected by schizophrenia with long duration of illness and multiple hospitalizations. Second, the overall discontinuation rate of depot antipsychotics at 12 months was 28.8%, with olanzapine and aripiprazole showing longer time to discontinuation compared with zuclopenthixol. Third, the type of prescribed LAI antipsychotic at baseline together with history of multiple substance use disorders may favor therapy discontinuation during the 12-month follow-up period. Finally, the prescription of a specific antipsychotic is influenced by patients’ characteristics.

In contrast with international clinical guidelines advocating for a broader and earlier use of LAI antipsychotics [30], our finding emphasizes that these formulations seem to be still underused by a significant proportion of practicing psychiatrists [31]. Nevertheless, this substantially expands the results of recent studies reporting that LAI antipsychotics are still mainly reserved to chronic patients, with multiple hospitalizations and long duration of illness [32]. Indeed, according to different surveys conducted to better understand the reasons of the low prescription of LAI formulations, the interviewees reported that LAIs persist to have an image problem, in particular in terms of coercion, exacerbated by the main use of these medications for the most stigmatized and chronically ill patients [33]. Our findings underline, therefore, the importance of an early initiation of LAI antipsychotics for patients affected by psychotic disorders in order to promote adherence to medications, to prevent relapses and rehospitalizations.

Even though the available literature shows contrasting data about the preferential prescribed compound among LAI antipsychotics [15,31], guidelines do not provide recommendations on which LAI should be selected, suggesting that the same criteria recommended for the choice of an oral antipsychotic be taken into account. Notably, haloperidol was found to represent the most frequently administered compound, in line with previous reports [34,35]. Taking into account real-world data from recent studies, to the best of our knowledge, this is the first study showing higher prescription of haloperidol decanoate as compared to other LAI antipsychotics. As has emerged from previous studies, the discrepancy in patterns of prescription could be explained by study methods, local factors such as refundability and socio-economic context, or by the progressive introduction of depot formulation for SGAs [15,31]. It is difficult to compare our data with the available literature, as differences in local clinical practice could influence the frequencies of LAI antipsychotic prescriptions [36]. However, nearly 30% of our sample discontinued LAI antipsychotics at 12 months. A previous research, albeit with shorter follow-up periods [37], suggests that discontinuation of LAI antipsychotics is common in patients with schizophrenia spectrum disorder. We hypothesize, therefore, that additional measures should be applied to maximize the benefits of LAIs. Clinicians should keep a therapeutic alliance and regularly discuss the factors that can hamper treatment continuation with their patients. Particularly, shared decision-making strategies may represent a mainstay in improving adherence to medication, and thus, in preventing discontinuation. Indeed, discontinuation of antipsychotic medication is associated with a high risk of relapse, hospitalization and mortality [38]. Non-adherence to treatment in schizophrenia and bipolar disorder is multifactorial. Nevertheless, many of these factors may be modifiable and can be specifically targeted in early intervention programs. In particular, a negative attitude, lack of insight, poor quality of life, cognitive impairment, negative symptoms, side effects of medications and alcohol or substance use disorders are associated with or are predictive of nonadherence to oral or LAI antipsychotics [39]. Nevertheless, one of the most important challenges in treating individuals with schizophrenia and related disorders is the ongoing functional disability related to negative symptoms, cognitive impairment and drug resistance after each acute episode. In this regard, negative symptoms including apathy, anhedonia, low motivation and diminished verbal and/or non-verbal expression can be present at the onset of illness and can represent the predominating symptoms. Although different studies report a lack of association or a very weak association between negative symptoms and insight as well as treatment adherence, negative symptoms are associated with greater disability and poor long-term prognosis [40]. Despite this, several measures to favor treatment continuation should be taken into account, including psychosocial interventions such as psychoeducational or supportive psychotherapy.

Overall, depot antipsychotics appear to be an effective treatment strategy for promoting adherence [8,12,13,41], although metanalyses of randomized controlled trials (RCT) comparing LAI versus oral antipsychotics found mixed results with regard to discontinuation rates [42]. Globally, the risk of relapse or rehospitalization is approximately 20% to 30% lower with LAI antipsychotics than equivalent oral formulations [12,38,43]. Different factors contribute to LAI antipsychotic continuation including type of compound, health policies and socio-economic context [28]. A very recent Italian, multicentre, prospective study, including 394 patients initiating a LAI antipsychotic found an overall discontinuation rate of 39.3% at 12 months, with risperidone-LAI and olanzapine-LAI showing the lowest and the highest discontinuation rates, respectively [44]. Yan et al. (2018) reported that about 75% of American patients with bipolar disorder and schizophrenia who initiated aripiprazole-LAI discontinued treatment after one year [45]. Moreover, an Austrian study by Rittmannsberger et al. (2017) found that nearly half of patients discontinued depot antipsychotics after 6 months [27]. On the other hand, two systematic reviews evaluated the discontinuation rate during long-term treatment with SGA-LAIs. In the first one, Saucedo Uribe et al. (2020) compared the long-term efficacy and safety of SGA-LAIs versus FGA-LAIs among adults with schizophrenia (treatment duration ≥12 weeks) [46]. Authors found no differences between FGA and SGA-LAIs for all causes of discontinuation and overall change in symptoms. Zuclopenthixol decanoate was the only FGA that was reported to be less efficacious than risperidone. Finally, the systematic review by Gentile (2019) reported that about 50% of patients with schizophrenia spectrum or bipolar disorders discontinued treatment with SGA-LAIs before 36 weeks [16]. Overall, similar rates of discontinuation were observed in studies on aripiprazole-LAI and olanzapine-LAI, whilst paliperidone and risperidone-LAI showed lower discontinuation rates in head-to-head studies. Therefore, even though controversial data are reported in literature [47], our results would suggest that some compounds (olanzapine and aripiprazole) can promote treatment continuation compared to others (zuclopenthixol). Thus far, it is still unclear whether SGA-LAIs may be more effective than FGA-LAIs in improving adherence of patients with affective and non-affective psychotic disorders [9], although the lowest risk of mortality was observed for SGA-LAIs, particularly for once-monthly paliperidone LAI and risperidone LAI [48]. Our finding of a shorter time to discontinuation for zuclopenthixol decanoate, compared with both olanzapine-LAI and aripiprazole-LAI, may be explained by some pragmatic reasons [44,49] such as the higher risk of extrapyramidal side effects (EPS) for FGAs versus SGAs [50]. In this regard, a recent network meta-analysis, including individuals with non-affective psychotic disorders [51], reported that aripiprazole-LAI was superior to other LAI antipsychotics with regard to acceptability. Furthermore, even though olanzapine-LAI was reported to have equivalent discontinuation rates than other LAI antipsychotics [12], it is possible that the longer time to discontinuation of olanzapine-LAI, as reported in our study, may reflect a more engagement of health professionals for post-injection monitoring, in turn favoring long-term adherence [20].

There are some obstacles in the way of achieving the theoretical advantages of LAIs on relapse prevention. Although the predictors of antipsychotic LAI discontinuation were not studied extensively, different factors including the number of hospitalizations before treatment, the setting where the treatment was started, dose of LAI antipsychotic, history of combined antipsychotic treatment, substance abuse and duration of illness were found to be related to treatment discontinuation [37,52] Similarly, we showed that the type of prescribed antipsychotic LAI at baseline together with a history of multiple substance use disorders may favor treatment discontinuation during the 12-month follow-up period. In this regard, some authors reported that early non-response to an antipsychotic is associated with long-term discontinuation of this antipsychotic [53]. For instance, a RCT study found that early response to olanzapine LAI predicts long-term clinical stabilization, similarly to what happens with oral antipsychotics [54]. On the contrary, the development of antipsychotic-related adverse effects is one of the reasons for early treatment discontinuation [44]. Comorbid substance use disorders may induce early interruption of antipsychotics [55] and this aspect is clinically relevant, as nearly 50% of individuals with schizophrenia or bipolar disorder have a lifetime history of substance use disorders [56,57]. However, multiple factors are potential mediators of the association between pre-onset poly-substance abuse and a shorter time to medication discontinuation. First of all, cognitive impairment is a core characteristic of psychotic disorders [58,59] and a contributor to poor response to antipsychotics [60]. Substance use disorders worsen performances in different cognitive domains including memory, attention, psychomotor speed, executive functioning [61] and insight, the latter directly influencing medication adherence [62]. Second, several studies indicate that patients who have chronic substance abuse show poor treatment response and early medication discontinuation than those without substance use disorders [10,63]. One reason is that substances can affect LAI antipsychotic plasma levels, hampering clinical stabilization [64]. However, it should be taken into account that patients with comorbid substance-use disorders more frequently receive a LAI, as this group of individuals is generally less adherent to medical prescriptions. Despite this frequent co-occurrence and its negative impact on the course of psychotic disorders, there were few studies assessing optimal therapeutic strategies for these patients. Substance use disorders can further worsen the cognitive abilities of patients affected by psychotic disorders so that specific strategies might have to be applied to improve treatment adherence. Some effective integrated interventions for patients with dual diagnosis may include group counseling with motivational enhancement, cognitive behavioral therapy, social skill training, psychoeducation, family interventions and residential programs for the most challenging patients [57]. Particularly, interventions could be tailored based on the patient’s needs.

Noteworthy, the results of the present study highlight that the prescription of a specific antipsychotic is influenced by patients’ characteristics. Namely, some clinical reasons may be associated with antipsychotic LAI choice. Haloperidol decanoate and generally FGA-LAIs are usually prescribed in chronic patients affected by schizophrenia. Haloperidol decanoate, apart from its possible disadvantage related to motor side effects, is still frequently prescribed due to its low cost and the associated long clinical experience [65]. Furthermore, FGA-LAIs may be beneficial for controlling some symptoms such as aggressiveness and disorganization due to prominent positive symptoms [66]. In addition, some studies reported that men with schizophrenia are more prone to commit severe acts of violence than women [67] and this would explain the preferential prescription of FGAs in men than women. Regarding the relation between duration of illness and preferential prescription of FGAs, chronic patients are likely to be treated with the oldest compounds probably due to the availability of these compounds in long-acting formulations at the onset of their disorder more than due to a loss of severity of negative symptoms with the progression of schizophrenia [36,68].

Another interesting result is that patients receiving SGA-LAIs are more likely to receive psychotherapy than those in treatment with FGA-LAIs. The relationship between psychotherapy and prescription of SGAs can be interpreted as the result that subjects receiving psychotherapy may have more prominent affective symptoms or on the contrary that SGAs are more beneficial on cognitive symptoms than FGAs [69]. Of note, a more frequent use of aripiprazole-LAIs in patients without a diagnosis of schizophrenia is in line with a previous study by Barbui et al. (2020) and may call forth the licensed indications of oral aripiprazole for individuals with bipolar disorder [20]. Nevertheless, SGA-LAIs are reported to be associated with a longer time to discontinuation and a lower use of concomitant agents to treat extrapyramidal side effects than FGA-LAIs among patients with bipolar disorder [4,70]. On the other hand, we found that SGA-LAIs are preferentially prescribed in individuals with pre-onset substance use disorders. Some authors reported that switching from other antipsychotics to aripiprazole may result in symptom improvement in patients with bipolar or schizoaffective disorders and comorbid substance misuse [71,72]. Furthermore, aripiprazole may reduce cocaine craving in patients with schizophrenia or bipolar disorder [73]. In this regard, an Italian prospective study [74] and a Spanish retrospective study [75], both involving psychotic patients with coexisting substance use disorders, suggested that aripiprazole-LAI may be useful in improving substance craving.

With regard to side effects, in line with most literature, a treatment with haloperidol decanoate and, in general, with FGA-LAIs is associated with motor side effects, whereas olanzapine-LAI causes mainly sedation and sexual dysfunction [46]. Even though we found non-compliance as the main reason for discontinuation, almost 30% of individuals (40/138) stopped LAI antipsychotics as a result of side effects. In the light of a longer time of discontinuation associated with olanzapine-LAIs, we may assume an important role of motor symptoms in causing antipsychotic discontinuation. EPS are related to secondary negative symptoms, worse cognitive performances, reduced treatment adherence and suicide [76]. Moreover, motor side effects often require additional treatment with anticholinergic agents, burdening patients with other adverse effects such as delirium, memory impairment, and autonomic nervous system dysfunctions [13]. Therefore, not simply the efficacy but also the tolerability of antipsychotic treatment might profoundly affect adherence to therapy and clinical response [64]. On the other hand, despite the sexual dysfunction and sedation reported by individuals in treatment with olanzapine-LAI, a positive benefit/risk ratio for olanzapine-LAI in the long term was reported by some authors [49,51]. Overall, a better tolerability and more effectiveness on specific psychopathological domains (e.g., negative or cognitive symptoms) remain unmet needs for an ideal pharmacotherapy of psychotic disorders.

Some evidence suggests that antipsychotic treatment may contribute to reducing the volume of cortical matter in schizophrenia and bipolar disorder, but this effect may be less evident for SGAs compared to FGAs [77]. Indeed, it was hypothesized that SGAs may exert a neuroprotective effect, mediated by different molecular mechanisms including a stronger binding to serotonin 5HT-2A than dopamine D2 receptors [78], and depending on the doses [79]. On the contrary, FGAs may exert neurotoxic effects at all doses, leading to neurodegeneration [80]. Therefore, SGAs should be considered as the preferred first-line antipsychotic therapy.

In order to translate these results into clinical practice, it should be investigated whether treatment discontinuation could be prevented by training clinicians on existing evidence or by providing information and education to the patients in a standardized way (e.g., by a short information sheet or a standardized interview). Second, from our point of view, the observed few prescriptions of SGA-LAIs may further stimulate the ongoing discussion whether the newer LAI antipsychotics should be the preferential choice in first-episode patients. Third, it would be relevant to develop LAI formulations for the other atypical antipsychotics.

The study presents different limitations related to its naturalistic, non-interventional design. First, we acknowledge that although time to discontinuation could be considered a pragmatic outcome measure, the lack of a randomized controlled design prevents us from drawing definitive conclusions related to the comparative efficacy of the different LAIs. Nevertheless, a strength of the present study includes the large number of patients (n = 382) with affective and non-affective psychotic disorders, enrolled in outpatient services and day hospital of two different Mental Health Departments (one in urban and the other in sub-urban environment), thus offering a comprehensive overview and complexity of a real-world clinical setting. Second, for some LAIs, including olanzapine and risperidone-LAI, the number of patients was relatively small, limiting the precision of our findings. Moreover, paliperidone 3-monthly LAI has recently become available in Italy. Third, rating scale scores were not collected to assess symptom severity and side effects. Moreover, no subjective questionnaires regarding satisfaction or quality of life with therapy were provided to patients, as reported in other studies [20]. Nevertheless, the observation period was limited to 12 months, whilst a longer period would have provided further information. Finally, some local factors including availability of antipsychotics, local guidelines and long-standing habits could have influenced the prescribing attitude in the two centers. In order to address these limitations and overcome possible sources of bias, further multicenter studies are required to confirm our findings. Furthermore, future research may have the objective to include comparisons between patients initiating a LAI with those who did not, in order to clearly detect which factors are crucial in the decision of starting LAI formulations as well as a specific antipsychotic.

Despite such limitations, we believe that the present study is important since it can provide insight into the practical consequences of starting different LAI antipsychotics in real world settings.

## 5. Conclusions

This study provides important information about predictors of discontinuation of LAI antipsychotic treatment and patterns of prescription. Approximately 30% of patients stopped pharmacotherapy soon after initiation of a LAI antipsychotic, mainly as a result of lack of compliance or side effects. However, some SGA-LAIs, including aripiprazole and olanzapine-LAIs, showed a longer time to discontinuation than zuclopenthixol depot. The type of prescribed LAI antipsychotic at baseline as well as pre-onset poly-substance abuse seems to affect treatment continuation of a LAI antipsychotic. Furthermore, our research showed that patients’ characteristics influence prescription patterns. The choice of a specific LAI antipsychotic needs to be carefully discussed with the patient, taking into account individual features and possible barriers. A regular and careful assessment of adherence levels before and in the course of LAI treatment could be helpful to predict and monitor the risk of discontinuation. Nevertheless, it should be taken into account that the first relapse is a strong predictor of a rapid progression of both bipolar disorder and schizophrenia [81] and that the long-term effectiveness of an antipsychotic may decrease considerably after the second relapse [82]. Thus, the prevention of the second relapse is fundamental, and all individuals should receive a sufficient antipsychotic dose together with psychoeducation to avoid re-exacerbations of illness [83]. Therefore, non-adherence is a priority public mental health problem despite treatment advances. However, increasing knowledge about factors related to non-adherence and leveraging novel technologies may enhance early assessment and adequate management of individuals with severe mental disorders [84]. In this regard, the use of SGAs (including aripiprazole, olanzapine and paliperidone) may be the best choice for the maintenance treatment of affective and non-affective psychotic disorders, considering that both oral and LAI formulations of these molecules have the highest evidence regarding relapse prevention and acceptability by the patients [49]. Large and high-quality head-to-head studies comparing the different LAI antipsychotics should be performed in order to overcome the methodological limitations mentioned above. Moreover, studies recruiting patients after the first episode of illness are needed to provide novel insights as regards the clinical utility of LAI antipsychotics when used in the early stages of the disorder, reversing the idea that LAI antipsychotics are a therapeutic option dedicated to patients with the most serious and chronic forms of psychotic disorders. Indeed, in the light of our findings concerning long duration of illness and number of previous hospitalizations, we hypothesize that the main reasons for LAI use seem to have remained essentially unchanged over the time [6]. Nevertheless, this observational study reports the contradictions and complexity of a real-world setting, as denoted by the large use of FGA-LAIs. The following statements would remark the significance of this observational retrospective study and its usefulness for future investigations on this topic: (1) the attitude of psychiatrists toward antipsychotic LAI formulations could strongly influence the acceptance of these treatments by the patients and the stigma related to these treatments; (2) the optimal practice in prescribing an antipsychotic LAI is to identify patients’ needs; (3) patients should be involved in the therapeutic process, even when the awareness of illness is precarious and therapeutic adherence is poor; (4) treatment continuation may be supported by standardized information procedures and psychosocial interventions.

## Figures and Tables

**Figure 1 biomedicines-11-00314-f001:**
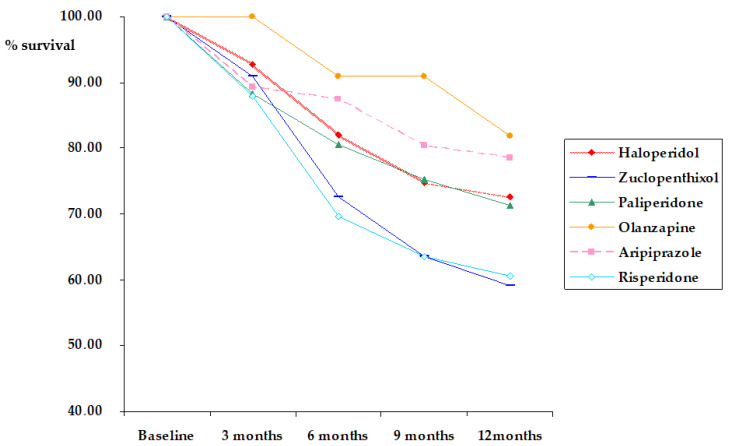
Survival in the different treatment groups. Zuclopenthixol versus Haloperidol: χ2 = 3.080. *p* = 0.079; Haloperidol: N = 150; Zuclopenthixol versus Paliperidone: χ2 = 1.506. *p* = 0.220; Zuclopenthixol: N = 44; Zuclopenthixol versus Olanzapine: χ2 = 4.358. *p* = 0.037; Paliperidone: N = 77; Zuclopenthixol versus Aripiprazole: χ2 = 3.877. *p* = 0.049; Olanzapine: N = 22; Zuclopenthixol versus Risperidone: χ2 = 0.004. *p* = 0.949; Aripiprazole: N = 56; Risperidone: N = 33.

**Table 1 biomedicines-11-00314-t001:** Descriptive analyses of the total sample.

Variables		Total Sample N = 382
**Gender**	Male	221 (57.9)
	Female	161 (42.1)
**Age (years)**		45.43 (12.79)
**Work status**	Employed *	304 (79.6)
	Unemployed	78 (20.4)
**Marital status**	Single	221 (57.9)
	Married/cohabitant	161 (42.1)
**Age at onset (years)**		27.04 (8.29)
**Diagnosis**	Bipolar Disorder	101 (26.4)
	Schizoaffective disorder	59 (15.4)
	Schizophrenia	222 (58.2)
**Duration of illness (years)**		18.52 (12.70)
**Duration of Untreated Illness (DUI) (years)**		2.60 (5.32)
**Presence of personality disorders**	Yes	52 (13.6)
	No	330 (86.4)
**Family history of psychiatric disorders** Missing: 10	Yes	144 (38.7)
	No	228 (61.3)
**Multiple family history of psychiatric disorders** Missing: 11	Yes	42 (11.3)
	No	329 (88.7)
**Pre-onset psychiatric comorbidity**	Yes	65 (17.0)
	No	317 (83.0)
**Pre-onset psychiatric poly-comorbidity**	Yes	3 (0.8)
	No	379 (99.2)
**Post-onset psychiatric comorbidity**	Yes	14 (3.7)
	No	368 (96.3)
**Post-onset psychiatric poly-comorbidity**	Yes	2 (0.5)
	No	380 (99.5)
**Pre-onset medical comorbidity**	Yes	63 (16.5)
	No	319 (83.5)
**Pre-onset medical poly-comorbidity**	Yes	13 (3.4)
	No	369 (96.6)
**Post-onset medical comorbidity**	Yes	158 (41.4)
	No	224 (58.6)
**Post-onset medical poly-comorbidity**	Yes	82 (21.5)
	No	300 (78.5)
**Pre-onset substance misuse**	Yes	80 (20.9)
	No	302 (79.1)
**Pre-onset poly-substance misuse**	Yes	41 (10.7)
	No	341 (89.3)
**Post-onset substance misuse**	Yes	86 (22.5)
	No	296 (77.5)
**Post-onset poly-substance misuse**	Yes	49 (12.8)
	No	333 (87.2)
**Presence of previous suicide attempts**	Yes	54 (14.1)
	No	328 (85.9)
**Number of previous suicide attempts**		0.22 (0.67)
**Presence of previous hospitalizations**	Yes	362 (94.8)
	No	20 (5.2)
**Number of previous hospitalizations**		4.60 (4.64)
**History of criminal acts**	Yes	44 (11.5)
	No	338 (88.5)
**LAI antipsychotic treatment**	Haloperidol decanoate	150 (39.3)
	Zuclopenthixol decanoate	44 (11.5)
	Paliperidone palmitate	77 (20.2)
	Olanzapine pamoate	22 (5.7)
	Aripiprazole	56 (14.7)
	Risperidone	33 (8.6)
**First/Second generation LAI antipsychotic treatment**	First generation	194 (50.8)
	Second generation	188 (49.2)
**Survival at 12 months**	Yes	272 (71.2)
	No	110 (28.8)
**Months of survival**	Haloperidol decanoate	10.24 (3.14)
	Zuclopenthixol decanoate	9.34 (3.66)
	Paliperidone palmitate	10.06 (3.43)
	Olanzapine pamoate	11.36 (1.76)
	Aripiprazole	10.55 (3.16)
	Risperidone	9.30 (3.72)
	Total	10.13 (3.28)
**Reason for discontinuation of LAI antipsychotic**	No discontinuation	244 (63.9)
	Recurrence (including hospitalization)	20 (5.2)
	Side effects	40 (10.5)
	No compliance	78 (20.4)
**Current poly-pharmacotherapy**	Yes	210 (55.0)
	No	172 (45.0)
**Treatment side effects**	Yes	150 (39.3)
	No	232 (60.7)
**Presence of multiple side effects**	Yes	35 (9.2)
	No	347 (90.8)
**Lifetime psychotherapy**	Yes	63 (16.5)
	No	319 (83.5)
**Type of lifetime psychotherapy**	None	319 (83.5)
	Psychoeducation/supportive	40 (10.5)
	Cognitive-Behavioral Therapy	16 (4.2)
	Psychodynamic	7 (1.8)
**Current psychotherapy**	Yes	9 (2.4)
	No	373 (97.6)
**Type of current psychotherapy**	None	373 (97.6)
	Psychoeducation/supportive	6 (1.6)
	Cognitive-Behavioral Therapy	2 (0.5)
	Psychodynamic	1 (0.3)

LAI = Long-Acting Injection; SD = Standard Deviation. N and percentages are reported for qualitative variables. Mean with SD are reported for quantitative variables. Percentages or SD are reported into brackets. * Employed people include students.

**Table 2 biomedicines-11-00314-t002:** Summary of Cox regression model.

Predictors	B	*p*	Exp(B)	CI
Age	0.204	0.520	1.226	0.659–2.279
Age at onset	−0.207	0.514	0.813	0.437–1.512
Duration of illness	−0.215	0.497	0.807	0.434–1.499
Duration of untreated illness	0.026	0.200	1.027	0.986–1.069
Presence of personality disorders (yes/no)	−0.057	0.871	0.944	0.473–1.884
Family history for psychiatric disorders (yes/no)	−0.051	0.835	0.950	0.588–1.536
Work status (employed versus the others)	0.023	0.929	1.023	0.619–1.691
Marital status (married/in partnership versus the others)	0.050	0.840	1.051	0.648–1.706
Pre-onset psychiatric comorbidity (yes/no)	0.191	0.549	1.210	0.648–2.259
Post-onset psychiatric comorbidity (yes/no)	0.184	0.764	1.202	0.360–4.013
Pre-onset substance use disorders (yes/no)	−0.073	0.862	0.930	0.411–2.105
Post-onset substance use disorders (yes/no)	0.193	0.640	1.213	0.540–2.724
Pre-onset medical comorbidity (yes/no)	−0.145	0.634	0.865	0.475–1.573
Post-onset medical comorbidity (yes/no)	−0.492	0.098	0.612	0.342–1.095
Diagnosis	NA	0.532	NA	NA
History of criminal acts (yes/no)	−0.332	0.290	0.718	0.388–1.328
Gender	0.191	0.463	0.463	0.727–2.014
Multiple family history of psychiatric disorders (yes/no)	−0.037	0.921	0.964	0.462–2.010
Pre-onset multiple substance use disorders (yes/no)	−1.049	0.058	0.350	0.118–1.035
Post-onset multiple substance use disorders (yes/no)	0.433	0.444	1.542	0.509–4.677
Pre-onset multiple medical comorbidity (yes/no)	−0.061	0.929	0.941	0.244–3.622
Post-onset multiple medical comorbidity (yes/no)	0.424	0.217	1.528	0.779–2.999
Type of LAI antipsychotic	NA	0.033	NA	NA
Poly-therapy (yes/no)	0.128	0.556	1.137	0.743–1.739
Lifetime psychotherapy (yes/no)	−0.404	0.158	0.668	0.381–1.169
Lifetime attempted suicide (yes/no)	−0.370	0.288	0.691	0.349–1.368
Lifetime hospitalizations (yes/no)	−0.108	0.812	0.897	0.369–2.185

NA = not applicable.

## Data Availability

Data available on request due to restrictions (e.g., privacy or ethical).

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
