# Peer review of "High Rate of Discontinuation during Long-Acting Injectable Antipsychotic Treatment in Patients with Psychotic Disorders"

_biomedicines, 2023, doi:10.3390/biomedicines11020314_

Round 1

Reviewer 1 Report

Dear Authors,

I have read the manuscript and I send you my comments:

Even if the type of the study is a limitation for this manuscript clinical data are necessary to clarify the answers to the aim of the study.

1) Please report the clinical difference of patients using each type of drug.

2) Please report the role of DDI in the difference of response in each group of treatment

3) please report the age related difference to the treatment

4) please report the correlation between ADRs in each group of patient considering also the role of DDI

5) In conclusion you say. "The results of the present study can help clinicians in the choice of LAI antipsychotic according to patients’ characteristics and in a context of precision medicine", but no data of clinical characteristic or of precision medicine have been reported   

6) Table1 must be described in caption. Moreover please separate clinical characteristic for each group of drugs and for each group of patient (age and gender)

7) Table 2 it is not necessary, you must report clinical characteristics of the patients more than statistical analysis

Author Response

Reviewer 1

Dear Authors,

I have read the manuscript and I send you my comments:

Even if the type of the study is a limitation for this manuscript clinical data are necessary to clarify the answers to the aim of the study.

First of all we would like to thank the first Reviewer for the useful suggestions to improve our manuscript.

1) Please report the clinical difference of patients using each type of drug.

Thanks for your request for clarification. The statistically significant clinical differences between drugs had been already reported  in the results, with the exception of pre-onset medical comorbidity. Specifically, the frequency of prescription of the compounds was different among diagnoses: patients affected by schizophrenia were more likely to receive haloperidol and less aripiprazole compared to subjects suffering from schizoaffective or psychotic bipolar disorder (χ2=59.943, p<0.001). The frequency of prescription of the compounds was different among genders (χ2=12.358, p=0.030): zuclopenthixol was preferentially prescribed in males than females (p<0.05). In addition , patients in treatment with paliperidone or olanzapine showed less pre-onset medical comorbidity than the other treatment groups (χ2=14.591, p=0.012). The distribution of treatments was also significantly different according to the lifetime administration of psychotherapy (χ2=12.287, p=0.031): patients who received psychotherapy during the course of life were more likely in treatment with aripiprazole or paliperidone (p<0.05). Finally, patients treated with zuclopenthixol had a significant longer duration of illness with respect to subjects in treatment with aripiprazole (p=0.029). In the last sentence of the results we specified that no other statistically significant clinical differences were found between treatment groups. Specifically no statistically significant differences were identified between treatment groups with regard to history of criminal acts (χ2=4.170, p=0.525), multiple family history of psychiatric disorders (χ2=5.375, p=0.372), pre-onset (χ2=6.228, p=0.285) and post-onset psychiatric comorbidity (χ2=8.396, p=0.136), pre-onset (χ2=1.790, p=0.877) and post-onset psychiatric poly-comorbidity (χ2=3.110, p=0.683), pre-onset (χ2=8.894, p=0.113) and post-onset poly-substance misuse (χ2=8.952, p=0.111), pre-onset medical poly-comorbidity (χ2=7.215, p=0.205) and post-onset medical poly-comorbidity (χ2=4.834, p=0.205), presence of previous suicide attempts (χ2=8.038, p=0.154), presence of previous hospitalizations (χ2=6.695, p=0.244), current poly-pharmacotherapy (χ2=4.850, p=0.434), type of lifetime psychotherapy (χ2=24.281, p=0.06), current psychotherapy (χ2=17.495, p=0.295), presence of personality disorders (χ2=6.515, p=0.259), work status (χ2=7.943, p=0.159), marital status (χ2=5.597, p=0.347), pre-onset (χ2=8.612, p=0.126) and post-onset substance misuse (χ2=10.208, p=0.07), post-onset medical comorbidity (χ2=1.903, p=0.862), age (F=1.583, 0.164), age at onset (F=1.393, p=0.226), duration of untreated illness (F=0.194, p=0.965), number of previous hospitalizations (F=1.582, p=0.164), number of previous suicidal attempts (F=0.856, p=0.551).

2) Please report the role of DDI in the difference of response in each group of treatment

Thanks for your suggestion. Patients in treatment with olanzapine showed a longer treatment duration compared to those in treatment with zuclopenthixol (p=0.019) and risperidone (p=0.012), as shown by univariate analysis weighted for duration of illness.  We reported this information in the results.

3) Please report the age related difference to the treatment

Thanks for your request for more information about age. No statistically significant differences were found between treatment groups regarding age.

In detail: haloperidol: 45.32 ± 12.491, zuclopenthixol: 49.73 ± 14.314, paliperidone: 44.06 ± 13.914, olanzapine: 42.18 ± 12.527, aripiprazole: 44.65 ± 11.277, risperidone: 46.88 ± 11.535.

4) Please report the correlation between ADRs in each group of patient considering also the role of DDI

Duration of illness was not significantly different in the two groups identified by the presence or absence of treatment side effects, even when co-variating for the type of compound (F=2.028, p=0.155). This information was reported in the text.

5) In conclusion you say. "The results of the present study can help clinicians in the choice of LAI antipsychotic according to patients’ characteristics and in a context of precision medicine", but no data of clinical characteristic or of precision medicine have been reported   

The data you requested has been provided. 

6) Table1 must be described in caption. Moreover please separate clinical characteristic for each group of drugs and for each group of patient (age and gender)

As you requested we reported more information about the clinical differences between treatment groups above.

7) Table 2 it is not necessary, you must report clinical characteristics of the patients more than statistical analysis

The data you requested has been provided. 

Reviewer 2 Report

This is a well-designed (and well written) study poised to identify the factors associated with LAI discontinuation in a real-world setting. To this end they analyzed a sample of 382 patients, of which one hundred and one subjects (26.4%) 136 were affected by psychotic bipolar disorder, 59 by schizoaffective disorder (15.5%) and 137 by schizophrenia (58.1%).

The authors found that the overall discontinuation rate of LAI antipsychotics at 12 months was 28.8%, with olanzapine and aripiprazole showing longer time to discontinuation compared with zuclopenthixol. The authors also found that some factors (including substance use) predicted LAI discontinuation.

I have a few comments that the authors may want to consider:

-       Inclusion/exclusion criteria include a treatment with a LAI antipsychotic < 2 months. This is quite a short period and might create censoring well before the first-time interval examined at 3 months. How many patients had only 3 months of treatment with LAI and why choosing this time frame?

-       The authors state that duration of untreated illness was considered as the time elapsing between the onset of the disorder and the prescription of a proper treatment (antipsychotic for schizophrenia and atypical antipsychotic or mood stabilizer for bipolar disorder). However, it is known that for most bipolar patients the first treatment is an antidepressant because the polarity of onset of the first episode is for 60/70% of patients depressive. Was this considered in the assessment of the duration of illness?

-       Still on duration of illness. The Cox proportional hazard model appears to include many predictors and some of them are redundant (i.e. illness duration is calculated via age, and so on) Was this considered and the authors performed separate analyses adjusting for age or duration of illness?

-       Was the assumption of proportionality of hazards controlled for?

-       It would be good to indicate in the figure how many patients are in each treatment group

-       Bipolar psychotic disorder is not really a diagnostic category, better maybe bipolar 1 disorder with psychotic features?

Minor points:

-       Sample size should be reported in the abstract

Author Response

Reviewer 2 

This is a well-designed (and well written) study poised to identify the factors associated with LAI discontinuation in a real-world setting. To this end they analyzed a sample of 382 patients, of which one hundred and one subjects (26.4%) 136 were affected by psychotic bipolar disorder, 59 by schizoaffective disorder (15.5%) and 137 by schizophrenia (58.1%).

The authors found that the overall discontinuation rate of LAI antipsychotics at 12 months was 28.8%, with olanzapine and aripiprazole showing longer time to discontinuation compared with zuclopenthixol. The authors also found that some factors (including substance use) predicted LAI discontinuation.

First of all, we would like to thank the reviewer for the interest in the manuscript and the useful suggestions aimed at improving the manuscript.

I have a few comments that the authors may want to consider:

- Inclusion/exclusion criteria include a treatment with a LAI antipsychotic < 2 months. This is quite a short period and might create censoring well before the first-time interval examined at 3 months. How many patients had only 3 months of treatment with LAI and why choosing this time frame?

Thanks for your right observation. We selected the time of 2 months to guarantee at least one follow-up visit. No patients were in treatment with 3-month paliperidone LAI. 14 patients had 2-month treatment (3.7%) and 20 had 3-month treatment (5.2%).

- The authors state that duration of untreated illness was considered as the time relapsing between the onset of the disorder and the prescription of a proper treatment (antipsychotic for schizophrenia and atypical antipsychotic or mood stabilizer for bipolar disorder). However, it is known that for most bipolar patients the first treatment is an antidepressant because the polarity of onset of the first episode is for 60/70% of patients depressive. Was this considered in the assessment of the duration of illness?

We are grateful for the important point the reviewer made. In the present study, antidepressant was considered as a possible treatment (also as initial treatment) in the management of bipolar disorder. Therefore and following your comment, we edited the Method section.

-  Still on duration of illness. The Cox proportional hazard model appears to include many predictors and some of them are redundant (i.e. illness duration is calculated via age, and so on) Was this considered and the authors performed separate analyses adjusting for age or duration of illness?

Thanks for your useful observations. We excluded from the model the variables that had a clear correlation such as the presence of lifetime suicide attempts and the number of lifetime suicide attempts. We performed the model removing duration of illness and the results were practically the same. The only statistically significant factors resulted in: type of treatment (p=0.037) and pre-onset poly-substance use disorders (p=0.051). We reported this information in the results.

- Was the assumption of proportionality of hazards controlled for?

Thanks for your consideration. We verified that age and duration of illness were not dependent by time, being these variables potentially influenced by this factor.  The proportion assumption hazard was satisfied for both these variables (age: p=0.07; duration of illness: p=0.444). We added this information in the text.

-  It would be good to indicate in the figure how many patients are in each treatment group

We provided this information in the figure.

-  Bipolar psychotic disorder is not really a diagnostic category, better maybe bipolar 1 disorder with psychotic features?

Thanks for your useful observation. We have changed it in the text.

Minor points:

-  Sample size should be reported in the abstract

Thanks for your right observation. The data has been added.

Reviewer 3 Report

The authors wanted to identify the factors associated with LAI discontinuation in a real-world setting by using a retrospective study design. They found that LAI discontinuation rate at 12 months was 28.8% and the type of baseline LAI antipsychotic predicted the probability of discontinuation, thus offering a comprehensive overview of the LAI use in a real-world clinical setting. The paper has the potential to contribute to the existing scientific literature on LAI use for patients with severe mental illness. I only have a few comments to further improve the quality of the authors’ paper. I have outlined these issues below:

1.

The introduction is well-written.

2.

When the authors compared the motor symptoms between each group of different LAIs, did they consider the CPZ equivalent doses or biperidin equivalent doses  between different LAI groups ?  

3.

The results are mostly consistent with results from previous research regarding the LAI use in patients with severe mental illness. There seems to be nothing novel for the professional readers. However, nothing novel may also be an important issue to be stressed in this article because the use of SGA-LAI nowadays does not have substantial progression as compared to the past. Some strategies to initiate SGA LAI will be helpful. For example, the share-decision making strategy for the patients and mental health care providers and the both having easy access to the opinions of advocates of SGA-LAI and the expert consensus for the LAI in treating schizophrenia patients.

4.

The SGA LAI seems to be underused from the results of this study. The readers may wonder if other real-world data from other research teams also show the similar trend. 

5.

Lack of insight, cognitive impairment, negative symptoms, side effects of medications, alcohol/substance use disorder all contribute to poor medication adherence to LAI  or oral antipsychotics in patients with psychosis. The author may talk more about the negative symptoms in the discussion.

6. Some issues about the neurotoxic effects of FGA LAI, neuro-protection effects of SGA LAI, the effects of FGA/SGA LAI on suicide and all-cause mortality, the latest data on real-world effectiveness of antipsychotic doses for relapse of schizophrenia can be briefly discussed. Please see the references as follows.

Nasrallah HA. Triple advantages of injectable long acting second generation antipsychotics: Relapse prevention, neuroprotection, and lower mortality. Schizophr Res. 2018 Jul;197:69-70. doi: 10.1016/j.schres.2018.02.004. Epub 2018 Mar 3. PMID: 29506767.

Taipale H, Mittendorfer-Rutz E, Alexanderson K, Majak M, Mehtälä J, Hoti F, Jedenius E, Enkusson D, Leval A, Sermon J, Tanskanen A, Tiihonen J. Antipsychotics and mortality in a nationwide cohort of 29,823 patients with schizophrenia. Schizophr Res. 2018 Jul;197:274-280. doi: 10.1016/j.schres.2017.12.010. Epub 2017 Dec 21. PMID: 29274734.

Vita A, De Peri L, Deste G, Barlati S, Sacchetti E. The Effect of Antipsychotic Treatment on Cortical Gray Matter Changes in Schizophrenia: Does the Class Matter? A Meta-analysis and Meta-regression of Longitudinal Magnetic Resonance Imaging Studies. Biol Psychiatry. 2015 Sep 15;78(6):403-12. doi: 10.1016/j.biopsych.2015.02.008. Epub 2015 Feb 16. PMID: 25802081.

Chen AT, Nasrallah HA. Neuroprotective effects of the second generation antipsychotics. Schizophr Res. 2019 Jun;208:1-7. doi: 10.1016/j.schres.2019.04.009. Epub 2019 Apr 11. PMID: 30982644.

Nasrallah HA, Chen AT. Multiple neurotoxic effects of haloperidol resulting in neuronal death. Ann Clin Psychiatry. 2017 Aug;29(3):195-202. PMID: 28738100.

Taipale H, Tanskanen A, Correll CU, Tiihonen J. Real-world effectiveness of antipsychotic doses for relapse prevention in patients with first-episode schizophrenia in Finland: a nationwide, register-based cohort study. Lancet Psychiatry. 2022 Apr;9(4):271-279. doi: 10.1016/S2215-0366(22)00015-3. Epub 2022 Feb 16. PMID: 35182475.

In the reviewer’s opinion, the above-mentioned issues need to be addressed by the authors.

Author Response

Reviewer 3

The authors wanted to identify the factors associated with LAI discontinuation in a real-world setting by using a retrospective study design. They found that LAI discontinuation rate at 12 months was 28.8% and the type of baseline LAI antipsychotic predicted the probability of discontinuation, thus offering a comprehensive overview of the LAI use in a real-world clinical setting. The paper has the potential to contribute to the existing scientific literature on LAI use for patients with severe mental illness.

We are glad the reviewer appreciated our manuscript.

 I only have a few comments to further improve the quality of the authors’ paper. I have outlined these issues below: 

  1. The introduction is well-written. 

Thank you for your positive comment.

  1. When the authors compared the motor symptoms between each group of different LAIs, did they consider the CPZ equivalent doses or biperidin equivalent doses  between different LAI groups ?   

Thanks for your consideration. The conversion of dose of LAI antipsychotic drugs into standard units (e.g. chlorpromazine equivalents) is currently an object of debate also due to the different administration period. Perhaps the formula with the most evidence is the one that uses olanzapine equivalents (doi: 10.1093/schbul/sbv167). If the antipsychotic doses are converted in olanzapine equivalents, the 4-week dose of zuclopenthixol resulted to be less than that of haloperidol (p<0.01), paliperidone (p<0.01) and olanzapine (p<0.01)with all the limits of this type of conversion. We reported this information in the results.

  1. The results are mostly consistent with results from previous research regarding the LAI use in patients with severe mental illness. There seems to be nothing novel for the professional readers. However, nothing novel may also be an important issue to be stressed in this article because the use of SGA-LAI nowadays does not have substantial progression as compared to the past. Some strategies to initiate SGA LAI will be helpful. For example, the share-decision making strategy for the patients and mental health care providers and the both having easy access to the opinions of advocates of SGA-LAI and the expert consensus for the LAI in treating schizophrenia patients. 

We agree with the Reviewer that some strategies to initiate SGA LAI may be helpful.Thus, we added some comments in the Conclusion section accordingly.

  1. The SGA LAI seems to be underused from the results of this study. The readers may wonder if other real-world data from other research teams also show the similar trend. 

Thank you for your question. Although some older studies have shown that SGA LAIs are underused (we reported in the manuscript Grover et al., 2019; Carr et al., 2016), to the best of our knowledge, no other report, apart from the present one, has indicated higher prescription of haloperidol decanoate as compared to other LAI antipsychotics. We added a sentence in the Discussion section.

  1. Lack of insight, cognitive impairment, negative symptoms, side effects of medications, alcohol/substance use disorder all contribute to poor medication adherence to LAI  or oral antipsychotics in patients with psychosis. The author may talk more about the negative symptoms in the discussion. 

We thank the Reviewer for this meaningful comment. We added some comments on this important issue in the Results section.

  1. Some issues about the neurotoxic effects of FGA LAI, neuro-protection effects of SGA LAI, the effects of FGA/SGA LAI on suicide and all-cause mortality, the latest data on real-world effectiveness of antipsychotic doses for relapse of schizophrenia can be briefly discussed. Please see the references as follows. 

Nasrallah HA. Triple advantages of injectable long acting second generation antipsychotics: Relapse prevention, neuroprotection, and lower mortality. Schizophr Res. 2018 Jul;197:69-70. doi: 10.1016/j.schres.2018.02.004. Epub 2018 Mar 3. PMID: 29506767.; Taipale H, Mittendorfer-Rutz E, Alexanderson K, Majak M, Mehtälä J, Hoti F, Jedenius E, Enkusson D, Leval A, Sermon J, Tanskanen A, Tiihonen J. Antipsychotics and mortality in a nationwide cohort of 29,823 patients with schizophrenia. Schizophr Res. 2018 Jul;197:274-280. doi: 10.1016/j.schres.2017.12.010. Epub 2017 Dec 21. PMID: 29274734.; Vita A, De Peri L, Deste G, Barlati S, Sacchetti E. The Effect of Antipsychotic Treatment on Cortical Gray Matter Changes in Schizophrenia: Does the Class Matter? A Meta-analysis and Meta-regression of Longitudinal Magnetic Resonance Imaging Studies. Biol Psychiatry. 2015 Sep 15;78(6):403-12. doi: 10.1016/j.biopsych.2015.02.008. Epub 2015 Feb 16. PMID: 25802081; Chen AT, Nasrallah HA. Neuroprotective effects of the second generation antipsychotics. Schizophr Res. 2019 Jun;208:1-7. doi: 10.1016/j.schres.2019.04.009. Epub 2019 Apr 11. PMID: 30982644;  Nasrallah HA, Chen AT. Multiple neurotoxic effects of haloperidol resulting in neuronal death. Ann Clin Psychiatry. 2017 Aug;29(3):195-202. PMID: 28738100.; Taipale H, Tanskanen A, Correll CU, Tiihonen J. Real-world effectiveness of antipsychotic doses for relapse prevention in patients with first-episode schizophrenia in Finland: a nationwide, register-based cohort study. Lancet Psychiatry. 2022 Apr;9(4):271-279. doi: 10.1016/S2215-0366(22)00015-3. Epub 2022 Feb 16. PMID: 35182475.

In the reviewer’s opinion, the above-mentioned issues need to be addressed by the authors.

Following the Reviewer’s remark and using the suggested references, we added some comments on these important issues in the Discussion and Conclusions section.

ACADEMIC EDITOR

The topic that the paper focuses on deserves the attention of selected reviewers and could be relevant to the readers' interest. I suggest the article continues through the reviewing process and possibly be accepted for publication provided the novelty and the statistical soundness of the reported results are assessed.

We would like to thank the Editor for the interest in the manuscript.

Round 2

Reviewer 1 Report

Dear Authors,

I have read the manuscript and I have seen that other data have been added, particularly in discussion. However no new data have been presented in order to clarify the answers to the aim of the study: " clinical and demographic predictors of LAI antipsychotic discontinuation ....; differences in the time of discontinuation .....; clinical factors (including tolerability) associated with the prescription ..."

No clinical data have been added in order to evaluate the characteristics of the patients using the drugs and also the characteristic of the patients with low adherence. No data related to the clinical characteristics of the patients with ADRs or DDIs have been added 

Author Response

Thanks for your request to provide additional clinical information. With regard to low-adherence, patients with poor compliance resulted to have a shorter duration of untreated illness (F=3.889, p=0.049), a lower number of lifetime suicide attempts (F=3.956, p=0.047), reported less multiple side effects (χ2=7.313, p=0.007), had a more frequent history of criminal acts (χ2=3.977, p=0.046) than the counterpart. No further statistically significant differences were identified.

With regard to adverse reactions, patients reporting side effects were older (F=5.458, p=0.020), more frequently women (χ2=7.353, p=0.007), more often treated with poly-therapy (χ2=12.131, p<0.001), with less frequent pre-onset medical poly-comorbidity (χ2=5.626, p=0.018), with less frequent pre-onset poly-substance use disorders (χ2=4.262, p=0.039) than the counterpart. No further statistically significant differences were identified.

This information has been added in the results.

Reviewer 2 Report

Excellent replies. No further comments.

Author Response

Excellent replies. No further comments.

Thanks for the appreciation of the manuscript.

Reviewer 3 Report

The authors address all the reviewer's comments carefully and have improved their manuscript considerably. I have no further comments.

Author Response

The authors address all the reviewer's comments carefully and have improved their manuscript considerably. I have no further comments.

Thanks for the appreciation of the manuscript.